# T-Lymphocyte Phenotypic and Mitochondrial Parameters as Markers of Incomplete Immune Restoration in People Living with HIV+ on Long-Term cART

**DOI:** 10.3390/biomedicines13081839

**Published:** 2025-07-28

**Authors:** Damian Vangelov, Radoslava Emilova, Yana Todorova, Nina Yancheva, Reneta Dimitrova, Lyubomira Grigorova, Ivailo Alexiev, Maria Nikolova

**Affiliations:** 1National Reference Laboratory of Immunology, National Center of Infectious and Parasitic Diseases (NCIPD), 1504 Sofia, Bulgaria; dvangelov@ncipd.org (D.V.); todorova_yana@ncipd.org (Y.T.); mstoimenova@ncipd.org (M.N.); 2Specialized Hospital for Active Treatment of Infectious and Parasitic Diseases, 1606 Sofia, Bulgaria; nyancheva@gmail.com; 3National Reference Confirmatory Laboratory of HIV, National Center of Infectious and Parasitic Diseases (NCIPD), 1504 Sofia, Bulgaria; naydenova.reneta@gmail.com (R.D.); lyubomiragrigorova@gmail.com (L.G.); ivoalexiev@yahoo.com (I.A.)

**Keywords:** HIV, antiretroviral treatment, immune restoration, viral suppression, immune exhaustion and senescence, mitochondrial dysfunction

## Abstract

**Background/Objectives**: Restored CD4 absolute counts (CD4AC) and CD4/CD8 ratio in the setting of continuous antiretroviral treatment (ART) do not exclude a low-level immune activation associated with HIV reservoirs, microbial translocation, or the side effects of ART itself, which accelerates the aging of people living with HIV (PLHIV). To delineate biomarkers of incomplete immune restoration in PLHIV on successful ART, we evaluated T-lymphocyte mitochondrial parameters in relation to phenotypic markers of immune exhaustion and senescence. **Methods**: PLHIV with sustained viral suppression, CD4AC > 500 and CD4/CD8 ratio >0.9 on ART (*n* = 39) were compared to age-matched ART-naïve donors (*n* = 27) and HIV(–) healthy controls (HC, *n* = 35). CD4 and CD8 differentiation and effector subsets (CCR7/CD45RA and CD27/CD28), activation, exhaustion, and senescence markers (CD38, CD39 Treg, CD57, TIGIT, and PD-1) were determined by flow cytometry. Mitochondrial mass (MM) and membrane potential (MMP) of CD8 and CD4 T cells were evaluated with MitoTracker Green and Red flow cytometry dyes. **Results**: ART+PLHIV differed from HC by increased CD4 TEMRA (5.3 (2.1–8.8) vs. 3.2 (1.6–4.4), *p* < 0.05), persistent TIGIT+CD57–CD27+CD28– CD8+ subset (53.9 (45.5–68.9) vs. 40.1 (26.7–58.5), *p* < 0.05), and expanding preapoptotic TIGIT–CD57+CD8+ effectors (9.2 (4.3–21.8) vs. 3.0 (1.5–7.3), *p* < 0.01) in correlation with increased CD8+ MMP (2527 (1675–4080) vs.1477 (1280–1691), *p* < 0.01). These aberrations were independent of age, time to ART, or ART duration, and were combined with increasing CD4 T cell MMP and MM. **Conclusions**: In spite of recovered CD4AC and CD4/CD8 ratio, the increased CD8+ MMP, combined with elevated markers of exhaustion and senescence in ART+PLHIV, signals a malfunction of the CD8 effector pool that may compromise viral reservoir latency.

## 1. Introduction

When applied early enough, contemporary antiretroviral therapy (ART) can prevent AIDS and significantly increase the life expectancy of people living with HIV (PLHIV) [1]. However, ART does not eliminate latent HIV viral reservoirs and does not prevent the low-level chronic immune activation associated with HIV reactivation, concomitant chronic infections, microbial translocation in the gut, or ART itself, thus finally leading to non-infectious degenerative diseases and accelerated aging [2,3].

CD4 0T cells are the principal target of HIV, and their absolute count (AC) has long served as a surrogate marker of viral activity used to determine the stage of immune deficiency, the right time to start ART, and the effect of antiviral treatment [4]. However, a number of studies have shown that restoring CD4AC above the critical level for immune deficiency (350 cells/µL) does not necessarily correspond to complete immune recovery [5].

Another classical immune parameter, the CD4/CD8 ratio, has proved to be a more sensitive indicator of on-going immune activation in the setting of long-term ART and suppressed HIV viral load [6]. CD4/CD8 ratio was shown to correlate with important biological and clinical parameters, such as the activation of the CD8 T cell pool, generally indicating active viral infection, the rate of non-infectious inflammatory co-morbidity (affecting the cardio-vascular, renal or nervous system), or the dimensions of the residual viral reservoir [2].

Further on, two phenomena are tightly linked to the process of continuous immune activation, and persistent HIV reservoirs: terminal differentiation and malfunction of virus-specific T cell clones expressing inhibitory receptors (immune exhaustion), and accelerated immune senescence (analogous to age-related immune senescence) characterized by altered subset composition, decreased thymic output, telomere shortening, and senescent-associated secretory phenotype of T cells [7]. The latter triggers mitochondrial dysfunction, resulting in increased ROS production and DNA damage, altogether driving the CD8 activation and systemic inflammation further. In this context, a number of phenotypic and functional markers were studied as potential biomarkers of incomplete immune restoration in response to ART, including the following: the expression of molecules associated with exhaustion like PD1, PD-L1, and TIGIT, inhibition (CD39+Treg; CD160), or apoptosis (CD57, FasL), the thymic output, relative telomere length (RTL), proliferative capacity, and glucose uptake or cytokine secretion potential of CD4 and CD8 T cells [8,9,10,11].

Mitochondria are key to immune cell functions by providing ATP via complete oxidative phosphorylation (OXPHOS), regulating calcium signaling and storage, as well as apoptosis signaling pathways [12]. Mitochondrial dysfunction leading to increased ROS production and DNA damage is a hallmark of aging, and therefore intrinsic to chronic HIV infection in the setting of continuous ART. Early initiation and strict adherence to ART have been shown to reduce the risk of comorbidities [13]. On the other hand, ART has also been associated with low-grade inflammation, mitochondrial dysfunction, and senescence in various cell types, even after relatively short-term exposure [14]. There is clearly a need to further investigate HIV- and ART-induced mitochondrial compromise and inflammaging, as well as to identify reliable parameters for monitoring the effects of long-term ART.

In the present study, we evaluated the mitochondrial mass and membrane potential of CD4 and CD8 T cells in relation to established phenotypic markers of immune exhaustion and senescence, as potential sensitive biomarkers of incomplete immune restoration in PLHIV on continuous ART and with undetectable HIV viral load.

## 2. Materials and Methods

### 2.1. Study Populations

Peripheral blood (5 mL) from PLHIV (*n* = 66) was collected at the Specialized Hospital for Active Treatment of Infectious and Parasitic Diseases, Sofia, Bulgaria, in the course of routine immune monitoring according to the national policies. Of them, 39 were on continuous ART (HIV+ART+), and 27 were ART-naïve (HIV+ART–). The ART+ individuals had been on therapy for least 2 years (range 2–24), with peripheral CD4AC above 500 cells/µL, and CD4/CD8 ratio > 0.9 during their last checkup. Control samples were obtained from HIV (–) age- and sex-matched individuals during prophylactic examinations (HC, *n* = 35). They were clinically healthy, without a history of chronic diseases or hospitalization in the last 2 months, or use of immunostimulatory or immunosuppressive drugs; no obesity (body mass index < 30); complete blood count and routine biochemistry tests within the normal ranges; no HSV, EBV, or active CMV infection, and no record of addictions or other dependencies.

Freshly obtained heparinized venous blood was employed for phenotyping analysis as well as for isolation of peripheral blood mononuclear cells (PBMCs) by density centrifugation using Lymphocyte Separation Medium (LSM, cat# LSM-A, Capricorn Scientific, Landkreis Marburg-Biedenkopf, Germany) for mitochondrial analysis.

The project was approved by the Ethics Committee of the National Center for Infectious and Parasitic Diseases—Sofia, Bulgaria. (Institutional Review Board/Institutional Ethics Committee (IRB/IEC) number: IRB 00006384, protocol No 3/2024).

### 2.2. Multicolor Phenotyping Analysis of Surface Markers

The share and AC of lymphocytes (CD45+), T cells (CD3+), helper T cells (CD3+CD4+), cytotoxic T cells (CD3+CD8+), proportions of CD4 and CD8 naïve (CD45RA+ CCR7+), central memory (CD45RA–CCR7+), effector memory (CD45RA–CCR7–), terminal effector memory (CD45RA+CCR7–), PD-1 expressed on CD4- and CD8-gated T cells, regulatory CD4 T cells (Treg, CD25^high^CD127–), and CD8 T cell effector subsets at different stages of differentiation/senescence/exhaustion (TIGIT and CD57 co-expression on CD27+28+; CD27+CD28–; CD27–CD28– subsets) were determined by standard multicolor flow cytometry.

For phenotypic analysis of CD4 and CD8 T cells surface markers, whole blood samples were stained by adding pre-titrated concentrations of directly conjugated antibodies suspended in staining buffer (cat# 342417, BD, San Jose, CA, US), followed by 15 min incubation at room temperature in the dark, 10 min incubation with fix-lysing solution (cat# 349202, BD), and repeated washing with PBS. CD4AC was determined in TRUCount tubes with MultiTest combination (cat# 342447, BD) according to the manufacturer’s instructions. For each test, 50 µL of whole blood was used. Stained samples were acquired immediately on a FACSCanto II flow cytometer using FACSDiva v. 6.1.3 software.

The following mAbs were used in the multicolor flow cytometry panel: anti-h CD3 AmCyan (cat# 339186), anti-h CD4 (PE cat# 565999), anti-h CD25 (APC-Cy7 cat# 557753), anti-hCD45RA (FITC cat# 555488), anti-h CCR7 (PE-Cy cat# 560765), anti-h CD38 (PE cat# 2117530), anti-h CD8 (APC cat# 340659), anti-h CD27 (AF700 cat# 356416), anti-h CD127 (PcpCy5.5 cat# 353220), anti-h CD8 (V450, cat# E-AB-F1110Q), anti-h CD57 (FITC cat# E-AV-F-1067C), anti-h CD45 (FITC cat# 2522025), anti-h CD45 (PerCP cat# E-AB-F1137F), anti-h PD-1 (APC cat#2496040), anti-h PDL-1 (PE cat# 2568040), anti-h CD28 (APC cat#377610), anti-h TIGIT (BV421 cat#2463550), and anti-h CD39 (cat #E-AB-F1165C). T-lymphocyte activation was evaluated by the number of CD38 molecules expressed on CD4+ and CD8+ T cells (CD38 antibody-binding sites, ABS) were quantified using the Quantibrite PE CD38 calibration flow cytometry kit (cat# 340495, BD Bioscience) according to manufacturer’s instructions.

Gating strategy: lymphocytes were defined on a CD45 vs. SSC plot; the proportions of basic T-lymphocyte subsets were determined after plotting CD3 vs. SSC, CD3 vs. CD4, and CD3 vs. CD8 expression, respectively. Naïve (CD45RA+CCR7+), central-memory (CM, CD45RA–CCR7+), effector memory (EM, CD45RA–CCR7–), and terminal effector memory (TEMRA, CD45RA+CCR7–) subsets were defined according to the combined expression of CD45RA vs. CCR7 within gated CD4+ and CD8+T cells. Tregs were defined as CD25^high^CD127– cells within gated CD4+ T lymphocytes. PD1 expression (both as percent positive cells and MFI) was determined on histograms within CD4- and CD8-gated T cells. TIGIT and CD57 co-expression was studied within CD4- and CD8-gated T cells, followed by a more detailed analysis within each of CD27/CD28-defined CD4 and CD8 T subsets.

### 2.3. Mitochondrial Analysis

Measurements of mitochondrial mass (MM) and mitochondrial membrane potential (MMP) were performed using MitoTracker Green FM for Flow Cytometry (MTG, cat #m46750) and MitoTracker Red (MTR, cat #m22425), respectively (Thermo Fisher Scientific, Waltham, MA, USA). PBMCs from whole blood were separated using density gradient centrifugation, transferred to 1 mL of colorless RPMI1640 + 10% Fetal Calf Serum, and stored at 4 °C overnight. Before staining, cells were allowed to reach room temperature. 1 × 10^5^ PBMCs were surface-stained with CD45, CD3, and CD8 mAbs to define CD4 and CD8 T cell subsets. Subsequently, 500 µL MTG and MTR in PBS were added at a final concentration of 1:2000 for MTG and 25 nM for MTR, respectively, incubated at 37 °C in a 5% CO_2_ humidified incubator for 30 min, washed once, and analyzed within 2 h. Fluorescence Minus One (FMO) controls were used for accurate gating of positive populations.

### 2.4. Statistical Analysis

Comparisons of quantitative parameters between groups were performed using the non-parametric Kruskal–Wallis test followed by post hoc Dunn test; correlations were evaluated by Spearman’s rank test. Statistical significance was set at *p* = 0.05. Quantitative variables were expressed as median and interquartile range (IQR); statistical analyses were carried out with SPSS v. 23 and GraphPad Prism v. 9.5.

## 3. Results

### 3.1. Basic Demographic and Immune Parameters of the Studied Groups

The basic demographic and laboratory characteristics of the patients’ and control groups are given in Table 1. No significant differences existed between age and sex distribution in the groups. According to the study design, ART+ PLHIV had suppressed HIV viral load (VL) and an immune response to continuous therapy. Consequently, CD4AC (median (IQR)) was significantly higher than that of ART– donors: 856 (711–1224) vs. 302 (136–527), *p* < 0.0001, and did not differ from that of HC 902 (784–1334), *p* > 0.05. In addition, the CD4/CD8 ratio of ART+HIV– donors was significantly higher as compared to ART–HIV+: 1.31 (1.13–1.71) vs. 0.31 (0.19–45), *p* > 0.001, and comparable to that of HC (1.8 (1.5–2.21), *p* > 0.05), precluding any on-going significant immune activation and inflammation process.

The evaluation of the subset composition of CD4 and CD8 T cell pools in terms of naïve, CM, EM, and TEMRA subsets showed a significant decrease in CM CD4, as well as in naïve and CM CD8 T in untreated HIV infection, at the expense of TEMRA CD4 and EM CD8, respectively (Figure 1).

ART completely restored the CD8 T cell subset composition, and almost completely that of CD4, with the exception of a slightly increased CD4 TEMRA subset: 5.3 (2.1–8.8) vs. 3.2 (1.6–4.4), *p* < 0.05.

### 3.2. Expression of Activation, Exhaustion, and Senescence Markers by CD4 and CD8 T Cells of PLHIV on Successful ART

We further evaluated well-accepted markers of CD4 and CD8 T cell chronic activation in HIV+ART+, as compared to untreated PLHIV and HC: the number of CD38 molecules (CD38 ABS) expressed by CD4 and CD8 T cells, and the share of total (CD25highCD127lo/neg), and induced (CD39+) CD4 Treg. Except for CD39+Treg, a significant increase was documented in untreated PLHIV, which returned to HC values after continuous and successful ART. Thus, CD38ABS on CD4 and CD8 T and percentage of total and CD39+Treg in HIV+ART+ were not significantly different from HC: 1291 (1002–1957) vs. 1021 (873–1379); 836 (666–1174) vs. 749 (626–865); 3.35 (2.55–4.73) vs. 2.7 (1.8–3.6); and 13.65 (3.25–33.93) vs. 18.1 (5.6–31.6), respectively (*p* > 0.05 for all).

PD1 is a well-characterized exhaustion marker expressed on T cells as a result of extreme stimulation. In line with this, we observed increased PD1 expression in untreated PLHIV that returned to HC values in the setting of ART: 329 (263–403) vs. 273 (213–328) vs. 267 (230–318), KW *p* > 0.05 and 320 (285–375) vs. 277 (226–303) vs. 210 (175–272), and KW *p* < 0.05 for CD4 and CD8 T cells of HIV+ART–, HIV+ART+, and HC, respectively. Likewise, TIGIT, another check point receptor associated with T cell exhaustion, was increased on CD4 and CD8 T cells of untreated patients, while not differing significantly from HC after successful ART: 35.7 (27.6–42.5) vs. 26.1 (18.7–30.5) vs. 21.7 (17.4–27.2) and 66.9 (55.3–79.1) vs. 52.9 (42.5–60.6) vs. 41.1 (32.4–56.2) for CD4 and CD8 T cells of HIV+ART–, HIV+ART+, and HC, respectively. (KW *p* < 0.05 for both, Appendix A).

Persistent HIV infection induces, in many ways, accelerated proliferation and telomere attrition of CD4 and CD8 T cells, and ART may aid to this effect. Well-accepted markers of viral driven exhaustion and premature senescence are TIGIT and CD57. The antigen-driven differentiation of effector T cells is characterized by the gradual loss of CD28 and CD27 co-stimulatory receptors, and acquisition of CD57 and/or TIGIT depending on the type of stimulus.

The comparison of these markers showed that even in the setting of successful ART, HIV+ART+ individuals exhibited significantly increased, but lower than HIV+ART–, CD27–CD28–CD8 T and CD57+TIGIT–CD8 T, subsets as compared to HC: 34.8 (28.5–55) vs. 57.8 (39.1–63.2) vs. 24.75 (8.67–33.5), *p* < 0.05, and 9.2 (4.3–21.8) vs. 12.5 (3.3–16.8) vs. 3.0 (1.5–7.3), *p* < 0.01 (Figure 2A,B).

To characterize these persistent changes in more detail, we studied CD57/TIGIT co-expression in each CD28/CD27-defined subset within the CD8 and CD4 pools. Changes were obvious in the setting of untreated HIV infection (Figure 3B), and most of them disappeared after treatment (Figure 3A,C).

The increased share of CD57+TIGIT– CD8 observed in ART+ PLHIV was mostly confined to the CD27–CD28– subset (Figure 3 and Figure 4A). At the same time, the increased expression of TIGIT on CD57–CD27+CD28– CD8 T in ART (–) PLHIV (Figure 3B) remained significantly higher compared to HC after treatment: 53.9 (45.5–68.9) vs. 74.1 (60–80.1) vs. 40.1 (26.7–58.5), *p* < 0.05 (Figure 3C and Figure 4A).

Notably, TIGIT– CD57+ subset correlated with the increase in effector CD8 T and was in inverse correlation with the TIGIT+CD57–CD27+CD28– intermediate effector subset (Figure 4B). The same subset analysis within CD4 T cell pool did not reveal any significant differences between ART+PLHIV and HC: TIGIT+CD57–CD27+CD28– CD4 39.1 (30.8–60) vs. 41.25 (23.83–50.8).

Neither TIGIT– CD57+ CD8 T nor TIGIT+ CD57– CD27+CD28– CD8 T cells correlated with the age or treatment duration in the group of ART+ PLHIV. On the other hand, TIGIT+CD57– CD8 subset correlated directly with CD4 and CD8 AC.

Thus, in PLHIV with suppressed HIV VL and restored CD4AC and CD4/CD8 ratio, we observed persistence of exhausted effector memory TIGIT+ CD8 T cells, at the expense of highly cytolytic CD57+ effectors within the intermediate pool. Therefore, the effector functions were largely confined to the apoptosis-prone CD57+CD27–CD28– subset.

### 3.3. Mitochondrial Function in CD4 and CD8 T Cells of PLHIV on Successful ART

Mitochondria are essential to T cell activity, and mitochondrial dysfunction is considered to precede T cell exhaustion. We evaluated the mitochondrial mass (MM) and mitochondrial membrane potential (MMP) in CD4 and CD8 T cells of ART+ PLHIV as key parameters of immune cell functionality, and compared them to treatment-naïve patients and HC. CD8 T cell MM was significantly increased in untreated PLHIV, and decreased in the setting of ART to the levels of HC: 3541 (1926–6443) vs. 2252 (1541–2583), *p* > 0.05 (Figure 5A).

On the other hand, CD4 MM remained comparable to ART (–) PLHIV: 7225 (4304–14504) vs. 7022 (6017–9884), *p* > 0.05), and significantly elevated as compared to HC: 7225 (4304–14504) vs. 4163 (3462–6073, *p* < 0.05) (Figure 5B). CD8 MMP was significantly increased in untreated HIV infection, and remained significantly higher in the setting of ART (ART+ vs. HC: 2527 (1675–4080) vs. 1477 (1280–1691) *p* < 0.01. (Figure 5C). Increased CD4 MMP did not reach a statistical significance in untreated PLHIV vs. HC: 3832 (2555–5576) vs. 2634 (2352–4388, *p* > 0.05). However, in spite of successful ART, it increased further and was significantly higher as compared to HC in ART+HIV+ donors: 5430 (3600–6534) vs. 2634 (2352–4388), *p* < 0.05 (Figure 5D). As expected, both mitochondrial parameters correlated closely with each other, and between CD 8 and CD4 T cells. Age impacted only MM but had no effect on MMP (Rho = 0.73; *p* < 0.0001 and Rho = 0.32; Rho = 0.35; *p* > 0.05, respectively). We did not find correlations between MM and MMP and CD4AC or CD4/CD8 nadirs, time from diagnosis to start of ART, or ART duration either (Figure 6A).

In addition, different ART regimens did not affect MMP and MM differently. However, we observed a significant correlation between the increased TIGIT– CD57+ CD8 subset and CD8 MMP in ART+ PLHIV (Rho = 0.45, *p* < 0.05) (Figure 6B).

## 4. Discussion

The concept of immune restoration, as a result of ART, has developed over the years, as well as the surrogate markers of “complete” immune recovery employed. It is now well accepted that CD4AC > 500 is not equivalent to restoration, and CD4/CD8 ratio is the more sensible predictor of non-infectious degenerative complication due to ongoing low-level immune activation. Moreover, CD4/CD8 ratio was proposed as a potential marker of HIV reservoir and its intermittent escape from suppression [6,15].

We studied a cohort of ART(+) PLHIV with both restored CD4AC and CD4/CD8 ratio in an attempt to identify biomarkers that were still significantly different from HC as a more subtle indicator of incomplete immune recovery. Our results showed that effector memory and effector CD8 T subsets expressing TIGIT and CD57 molecules, as well as CD8T MMP, remain significantly elevated in successfully treated patients, and that elevation was independent of age, therapy duration, time to treatment or CD4AC nadir. Most importantly, the increased level of senescent and exhausted CD8 T cells was combined with an increasing CD4 T cell MMP, despite the lack of important phenotypic aberrations within the T cell pool.

T cell immunoglobulin and ITIM domain (TIGIT) were identified as inhibitory immune check point, which is induced upon activation on NK and memory T cell subsets, and interferes with T cell metabolism, thereby affecting adaptive immunity. Expression of TIGIT within the CD8 T cell effector pool has been documented in different pathological settings [16]. HIV infection was shown to induce TIGIT, alongside with PD1, particularly on the intermediately differentiated CD27+CD8 effector pool containing also the HIV-specific effector cells. While TIGIT expression on CD4 T correlated with HIV viral load, this expression was retained on CD8 T effectors even in the setting of ART-controlled viral suppression. Simultaneous blocking of PD-L1 and TIGIT restored ex vivo effector T cell response in patients with advanced melanoma, and notably in chronic HIV infection, proposing TIGIT as a novel target for personalized therapy approach in PLHIV [17,18,19].

In line with, and extending the previous findings, we demonstrated persistently enhanced TIGIT expression in long-term treated PLHIV with stable HIV viral suppression (for at least 2 years), restored CD4AC and CD4/CD8 ratio, and no clinical or laboratory signs of immune activation. The elevated TIGIT expression affected the “intermediately differentiated “CD28–CD27+CD57– CD8 T subset, which has both cytolytic and proliferative potential, thus rendering a large fraction of viral-specific CD8+ T cells, including most of the HIV- and CMV-specific clones vulnerable to negative regulation. Persistent TIGIT expression in the setting of ART could be, but not only, a result of previous reactivations of HIV reservoirs and /or CMV infection. It should be noted that in our study group, TIGIT+CD8 subset did not correlate with expression of other activation or exhaustion markers (CD38, CD39, PD1), pointing rather to past than to on-going stimuli. One interesting observation was the direct correlation of TIGIT+ effector subset with CD4AC and CD8 AC, which supports the idea of virus-driven stimulation. Notefully, the so-called transitional memory (CD27+CD45RA–CCR7–) T cells that largely overlap with CD27+CD57–CD28– subset were shown to be the principal harbor of HIV reservoir [20].

We did not observe any correlation of increased TIGIT+ CD8 T cells with impaired mitochondrial parameters. However, TIGIT+ CD8+ T cells were shown to have significantly reduced expression of glycolysis genes, including *GLUT1*, *HK1*, and *HK2*, which resulted in impaired glucose uptake and glycolysis—defects that directly affected T cell functions in HIV infection [21]. Whatever the cause, persistent elevated TIGIT+CD57+CD8 T in ART+ PLHIV should be considered as an important possibility to improve immune restoration, and a target of specifically tailored immune therapy.

CD57 is a well-known biomarker of end-stage CD27(–) effector T cells, exhibiting high cytolytic activity but restricted proliferative potential [22,23]. In HIV progressors, uncoordinated upregulation of CD57, and accumulation of HIV-specific CD8+CD27+CD57+ cells were documented as a sign of impaired CD8 T cell differentiation contributing to failure of cellular immune control [24]. Lee et al. described the presence of CD57+ T lymphocytes in acute HIV infection that could be reversed with early-initiated treatment. Recently, a study by Elias Junior et al. demonstrated a higher proportion of cells expressing CD57 among ART(+) PLHIV with CD4AC > 350 as compared to age-matched HIV– controls, concluding that the restoration of normal levels of CD57-expressing T lymphocytes remains insufficient with current treatments [11]. We further extend this observation to (ART+) PLHIV with CD4AC > 500 and, importantly, with normalized CD4/CD8 ratio. The latter excludes any major immune activation processes of infectious, autoimmune or neoplastic origin. Indeed, there were no laboratory or clinical data of concomitant infections or chronic degenerative diseases among our ART+HIV+ study group.

It should be noted that the biological relevance of CD57 expression on CD8 effector cells is not unequivocal. Several studies have correlated the presence of CD57+ T lymphocytes with reduced immune performance typical of exhausted cells in other infections with *cytomegalovirus* [25], *Epstein–Barr* virus [26], *Mycobacterium tuberculosis* [27], and *Trypanosoma cruzi* [28,29]. On the other hand, an increased CD57+CD27+ CD8 T subset among melanoma TIL could proliferate in response to IL-2 and differentiate into CD57+CD27– T with increased perforin expression and potent anti-tumor cytotoxicity. Therefore, CD57 was defined rather as a marker of truly end-stage effector CTL, than of T cell senescence [23]. Also, CD57+ CD4 T cells were shown to exhibit significant cytotoxic activity against leukemia cells [30]. In our cohort of long-term treated patients with immune response to ART, CD57 expression was elevated mostly on the CD8 subset as compared to HC (Figure 2), and correlated inversely with the TIGIT+CD57– intermediate subset (Figure 4B). We could hypothesize an on-going stimulation of microbial or non-infectious origin, driving the differentiation of a limited number of TIGIT– clones to a terminal CD57+ stage, risking exhaustion in case of chronic or too vigorous activation.

The increased metabolic requirements of CD57+ effectors in our ART+ study group were confirmed by changes in T cell mitochondrial parameters. Mitochondria are essential for the vigorous metabolism of immune cells. Their compromise leads to increased oxidative stress, ROS accumulation, decreased membrane potential, and ultimately—to cellular apoptosis. It is well known that both HIV-infection and ART-treatment can contribute to mitochondrial damage, and hence—to accelerated senescence, and cellular dysfunction in PLHIV [14]. We examined MM and MMP of CD4 and CD8 T cells in PLHIV with completely restored CD4AC and CD4/CD8, aiming to evaluate the “price” of this formal restoration in terms of compromised metabolism, and its association with the observed aberrations in the effector cell pool. In line with the results of Yu, F et al. on large cohorts of ART-naïve and virally suppressed PLHIV, we observed increased CD4 and CD8 MM in active HIV infection as a sign of increased metabolic activity of virally stimulated immune cells [31]. On the other hand, in our ART+ cohort, CD8 MM returned to HC values, while CD4MM further increased to become significantly different from HC. Yu, F et al. explained the late MM rebound in both CD4 and CD8 T of ART+ PLHIV with accumulating effects of ART after more than 3 years of treatment [31]. In our cohort, however, there was no correlation between MM and ART duration, and most donors were on ART for a lot longer than 3 years (8y, 5–10). Moreover, the effects observed on CD4 and CD8 T cells were different. It is known that CD4 and CD8 T cells have different metabolic capacities and susceptibility to aging. A higher MM was described in CD4 TEMRA as compared to CD8 TEMRA, corresponding to a slower rate of immunosenescence [32]. It is tempting to speculate that the higher CD4 MM in the case of our ART+ cohort was associated with the observed significant enrichment in TEMRA CD4 T cells, and that was not a necessarily negative sign. However, a previous study by Masson et al. showed that CM CD4 T had the highest MM, comparable in HIV(–) and HIV+ART+ donors, and that CD4 CM T are a consistent part of HIV reservoirs [33]. Therefore, an increasing CD4 T MM in ART+PLHIV would be rather an alarming sign of HIV reservoir reactivation. Moreover, a significantly higher mitochondrial mass (MM) was reported in HIV-specific CD8+ T cells sensitive to apoptosis [34].

Cellular respiration and ATP synthesis depend largely on MMP. It was shown that changes in MMP play a key role in apoptotic cascade, and apoptosis is key to immune cells in the pathogenesis and progression of HIV infection. In our study we found an increased MMP of CD4 and CD8 T cells of PLHIV, that was not restored by ART. Moreover, MMP increased further in CD4 T of ART+ patients.

According to Sternfeld et al. MMP was reduced in HIV-infected, ART-naïve patients compared to HIV(–) healthy subjects, and correlated negatively with the decreasing CD4AC and the increasing percentage of apoptotic lymphocytes [35]. Another study in ART-naïve and ART-exposed PLHIV demonstrated a significantly higher level of CD8 MMP in ART-naïve patients compared with HIV-negative controls, while no significant difference in CD4 MMP was observed. After long enough ART (>3 years), MMP in CD8+T cells gradually recovered to nearly normal levels [31].

One explanation of the conflicting results on MMP in HIV patients might be the staining techniques. A comparison between tetramethylrhodamine methyl ester and different MitoTrackers showed that probes have different sensitivity to FCCP-induced depolarization and different numerical value expression [36]. Controversies may also be attributed to different characteristics of the study groups, as well as to the particular dynamics of this parameter. Thus, a decrease in MMP has been hypothesized to be a marker of apoptotic cells, including activated T lymphocytes. However, Matarrese P et al. demonstrated that the mitochondrial membrane was hyperpolarized once lymphocytes were activated, and this event preceded susceptibility to apoptotic cell death [37]. In addition, apoptosis proneness associated with an increased MMP could be exogenously modulated, e.g., by cytokines [38,39]. In line with those results, we propose that the increased CD8 T cell MMP observed in ART+PLHIV is an early apoptotic event, associated with continuous low-level activation that drives the gradual differentiation of the intermediate effector pool into terminal CD57+effectors, and, consequently, to its gradual exhaustion. A weakened capacity for CD8 effector response potentially increases the risk for reactivation of CD4 T cells, including those harboring dormant HIV. The significant correlation between increased MMP and CD57+TIGIT–CD57 CD8 T subset in ART+ PLHIV corroborates with this hypothesis.

It should be noted that prolonged exposure to ART was also shown to increase the adverse effects of HIV infection on mitochondrial functions [14]. Different ARV classes were shown to impact different T cell metabolism, with NRTIs, NNRTIs, and PIs but not INSTIs decreasing MMP [40]. We acknowledge, as a limitation, the small number of samples that were simultaneously phenotyped and assessed for mitochondrial markers in our study, and therefore, the impossibility to precisely evaluate ARV-specific adverse effects on MMP. Nevertheless, we did not observe any correlation between increased MMP and the following: ART regimen (INSTI vs. non-INSTI), ART duration, age, time to diagnosis, or time from diagnosis to treatment.

## 5. Conclusions

Our study contributes to the understanding of incomplete immune restoration in long-term ART-treated PLHIV by indicating T cell parameters that differ significantly from HIV– controls in spite of recovered CD4AC and CD4/CD8 ratio, and undetectable HIV VL. Increased T cell MMP is an early apoptotic event associated with continuous low-level activation. Persistent TIGIT+CD57– intermediate CD8 T subset indicates inefficient effector cell differentiation. Both parameters require attentive monitoring, and, if needed, targeted therapeutic intervention to prevent reactivation of latent viral reservoirs.

## Figures and Tables

**Figure 1 biomedicines-13-01839-f001:**
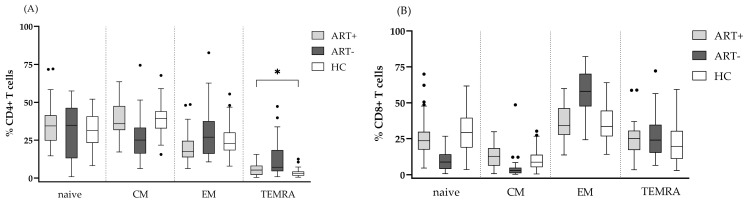
Comparison between the differentiation profiles of CD4 and CD8 T cell pools of the studied groups. The relative shares of naïve, central memory (CM), effector memory (EM), and terminal effector memory (TEMRA) subsets among CD4 T (**A**) and CD8 T (**B**) cells are presented. Statistical differences were evaluated by the non-parametric Kruskal–Wallis test. Only the statistical differences between ART+HIV+ and HC are marked, * *p* < 0.05. (•) represents outliers according to Tukey’s visualization. (HIV+ART+ are represented as light gray, HIV+ART– as dark gray, and HC as white).

**Figure 2 biomedicines-13-01839-f002:**
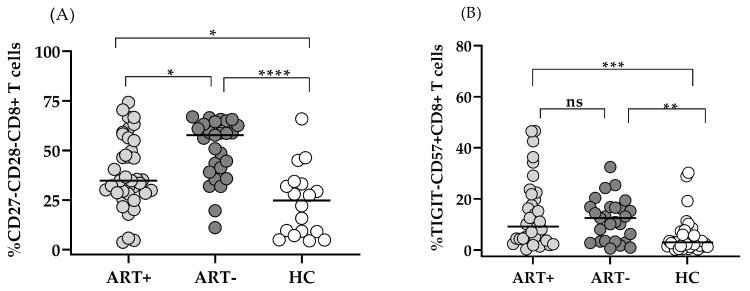
Effector CD8 T cell subsets with exhausted/senescent phenotype in the setting of ART: CD27–CD28– (**A**) and TIGIT–CD57+ (**B**). Individual values for the three studied groups are presented. Gating strategy: for A—the percentage of CD27–CD28– within CD8+CD3+-gated lymphocytes was determined; for B—TIGIT–CD57+ cells were determined as a percentage of CD8+CD3+-gated lymphocytes (* *p* < 0.05; ** *p* < 0.01; *** *p* < 0.001; **** *p* < 0.0001; ns–non-significant).

**Figure 3 biomedicines-13-01839-f003:**
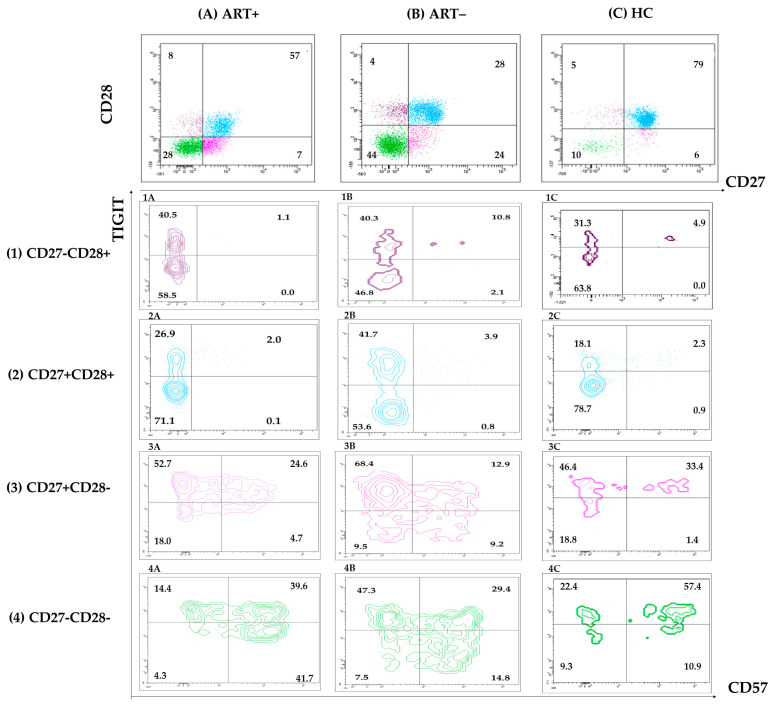
Co-expression of CD57/TIGIT on CD27/CD28-defined subsets of HIV+ART+ (**A**), HIV+ART– (**B**) and HIV– donors (**C**): samples were stained with a combination of CD57/CD28/CD8/CD3/TIGIT/CD27 mAbs. Gating strategy: Within the CD3+CD8+ gate, effector T cell subsets were determined by CD28/CD27 co-expression. The percentage of TIGIT+CD57–, TIGIT+CD57+, TIGIT–CD57+, and TIGIT–CD57– cells were determined in each CD27/CD28-defined CD8 T cell subset. Representative samples are shown.

**Figure 4 biomedicines-13-01839-f004:**
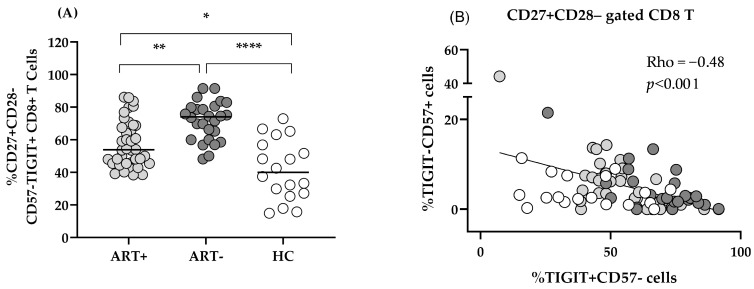
Exhausted CD8 T cells persist in the setting of successful ART: CD57–TIGIT+ CD27+CD28– CD8 T cells are compared between the studied groups (* *p* < 0.05; ** *p* < 0.01; **** *p* < 0.001; Kruskal–Wallis) (**A**). Inverse correlation between TIGIT+CD57–CD27+CD28– CD8 T and CD57+TIGIT– CD8 T in ART– (**B**). Individual values for the three studied groups are presented as follows: ART+ (light gray), ART– (dark gray), and HC (white); Spearman’s correlation test.

**Figure 5 biomedicines-13-01839-f005:**
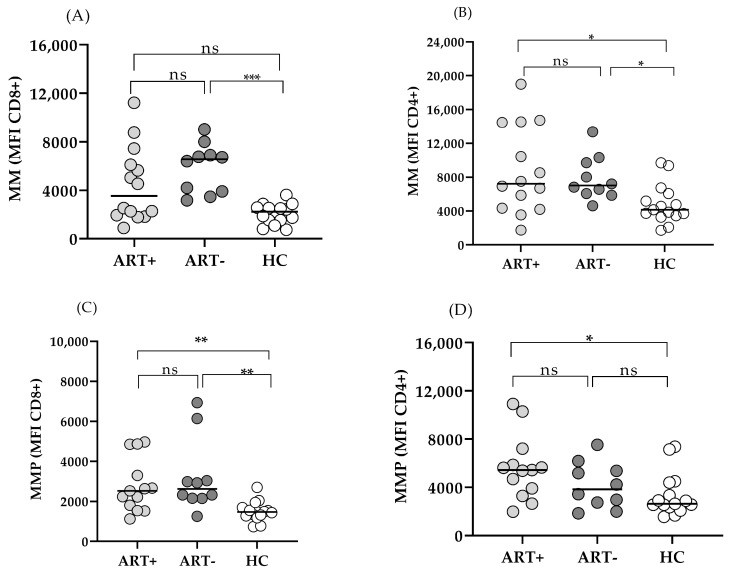
Mitochondrial parameters of T cells in the setting of ART. Mitochondrial mass (MM) MFI (**A**,**B**) and MMP (**C**,**D**) in CD8 (**A**,**C**) and CD4 (**B**,**D**) T cells of ART+ PLHIV, ART– PLHIV, and HC. Individual values are presented. * *p* < 0.05; ** *p* < 0.01; *** *p* < 0.001, ns–non-significant.

**Figure 6 biomedicines-13-01839-f006:**
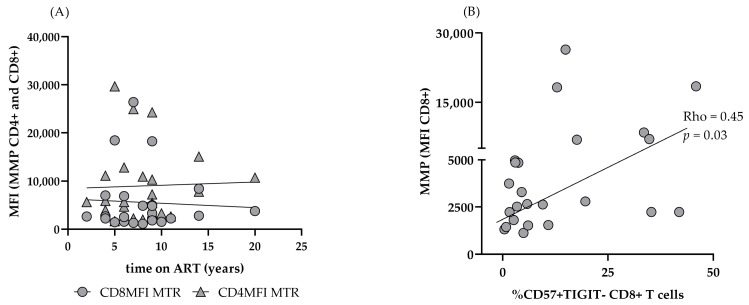
Correlations of T cell MMP in ART+ PLHIV. CD4 and CD8 T MMP does not correlate with ART duration (**A**); CD8 MMP correlates with CD57+ TIGIT– CD8+ T effectors (**B**).

**Table 1 biomedicines-13-01839-t001:** Basic demographic and immune characteristics of the studied groups.

Characteristics	HIV+ART+	HIV+ART–	HC	p^1^	p^2^	p^3^
Male sex (n)	26	22	21	ns	ns	ns
Female sex (n)	13	5	14	ns	ns	ns
Age (years)	45 (34–51)	40 (32–48)	37 (29–45.5)	ns	ns	ns
Time on ART (years)	8 (5–10)	na	na	-	-	-
Diagnosis to treatment time (years)	1 (0–5)	na	na	-	-	-
CD4 T AC (Cells/µL)	856 (711–1224)	302 (136–527)	902 (784–1334)	***	ns	***
Baseline CD4 T (cells/ µL)	511 (344–678)	339 (163–529)	na	*	-	-
CD8 T AC (Cells/ µL)	702 (528–878)	965 (441–1221)	505 (405–731)	ns	ns	**
CD4/CD8 ratio	1.31 (1.13–1.71)	0.31 (0.19–45)	1.8 (1.5–2.21)	***	ns	***

p^1^—HIV+ART+ vs. HIV+ART–; p^2^—HIV+ART+ vs. HC; p^3^—HIV+ART– vs. HC; quantitative variables are expressed as median (IQR); ns—non-significant, * *p* < 0.05, ** *p* < 0.01, and *** *p* < 0.001. Comparisons were performed using the chi square test for categorical and Kruskal–Wallis test for quantitative variables.

## Data Availability

The authors confirm that the data supporting the findings of this study are available within the article.

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
