# Peer review of "T-Lymphocyte Phenotypic and Mitochondrial Parameters as Markers of Incomplete Immune Restoration in People Living with HIV+ on Long-Term cART"

_biomedicines, 2025, doi:10.3390/biomedicines13081839_

Round 1

Reviewer 1 Report

Comments and Suggestions for Authors

Despite antiviral therapy, the incomplete immune restoration or reconstitution remained a hot topic in the field and has been linked previously to higher rate of mortality and morbidity. Authors attempted to explain the gap in literature regarding incomplete immune restoration in HIV patients by comparing the HIV+ART+, HIV+ART- and HIV-HC. Most of the studied parameters are already established/studied in HIV patients.

Few observations below:

  • Line 93: How much peripheral blood was collected.
  • Line 143: CM is missing in the CD45RA-CCR7+ subset. It may be written as (CM, CD45RA-CCR7+) like other subsets.
  • Figure 1 legend missing and it is difficult to understand figure. Add figure legends with group names HIV+ART+, HIV+ART-,HC.
  • Figure 1 start (*) represents significance that is present on one bar while other bars have multiple (•) that make the figure confusing about statistical analysis. Explain about these (•) in figure legend at least that what they represents?
  • Figure 1 x-axis legends may be write down one in figure legend to make figure clear and readers do not have to go back to method section for their explanation.
  • Line 220 use the word group A while it is not mentioned before what group A represents. Either write it down clearly in method section about groups and use same terminology at all places or write down what group A is in brackets.
  • Line 220 showed results that “group A (ART+) exhibited significantly increased CD27-CD28-CD8T and CD57+TIGIT-CD8 T subsets as compared to control”. As authors buildup hypothesis that incomplete immune restoration occurs in ART+ patients so it may be better to tell the difference of ART+ and ART- as well to clearly show that ART+ has middle range of immune restoration. The modified and more detailed statement will corresponds more clearly to the results represents at line 228 – 231.
  • Figure 3: Authors change the sequence of group A now and put ART- first and ART+ second place. This is not consistent with previous figures and grouping mentioned in results. I recommend to put ART+ data at first place being group A in study for better understanding.
  • Figure 3: for better understanding, percentages in dot plots will be more helpful to see the differences of different cell populations in all three groups and readers can compare them directly by looking at figure. OR et the end of each line of dotblot, small graph of percentages of same line will be helpful to compare.
  • Figure 4 a represents CD27+CD28- subset but it is missing in figure. So it is difficult to understand directly from figure which subset authors are talking about. I recommend to write down the CD27+CD28- subset on the top of graph OR y-axis legend
  • Mostly with incomplete immune restoration, CD4 T cells are discussed as well but authors only mention CD8 subset except line 229 where authors mentioned to study CD4 pools as well. Data is missing.
  • Figure 4B showed the individual values for three studied groups while from figure 4B, it is difficult to differentiate or understand groups subdivision.
  • Line 385, CD4 EMRA is used. While previously TEMRA is used. Same mistake at line 388.

Reviewer 2 Report

Comments and Suggestions for Authors
  1. What criteria were used to select and match the control and patient groups, particularly the healthy control group? Were potential confounding factors such as age, sex, comorbidities, or latent infections (e.g., CMV) considered during group selection?
  2. The captions and explanations for the figures and tables should be revised for greater clarity and completeness, ensuring they are self-explanatory and provide sufficient context for interpretation without referring back to the main text.
  3. For some cell populations, it is necessary to include a detailed gating strategy. Providing additional information or a supplementary figure illustrating how exhaustion and senescence markers (e.g., TIGIT, CD57, PD-1) were identified would enhance the clarity and reproducibility of the flow cytometry analysis.
  4. Table 1 needs substantial revision, as it is currently too basic and lacks clarity. The use of vague subtitles such as 'Parameters' does not provide meaningful context. It would be helpful to organize the data with clear, descriptive headings for each parameter and group (e.g., HIV+ART+, HIV+ART−, HC), and to explicitly define the p-values (e.g., p1, p2, p3) and the comparisons they refer to.
  5. The conclusion summarizes the key findings, but it would benefit from clearer language and improved structure. I recommend revising it to enhance readability and ensure that the main messages are effectively communicated. Specifically, the authors should: Clearly state how the findings contribute to the understanding of incomplete immune restoration in ART-treated PLHIV. Emphasize the clinical relevance of elevated CD8+ MMP as a potential biomarker. Clarify the therapeutic implications of persistent TIGIT+ intermediate effectors.

    A more concise and polished conclusion would strengthen the overall impact of the manuscript.

Reviewer 3 Report

Comments and Suggestions for Authors

Authors show in this research article that HIV+ individuals on long-term effective ART with CD4 counts >500 and near normal/normal CD4/CD8 T cell ratio show increased mitochondrial membrane potential in CD8 T cells and elevated makers of exhaustion and senescence as compared with HIV-ve healthy controls. The study reinforces the notion that HIV+ART+ individuals despite CD4 counts >500/cmm should not be considered as a sign for immune restoration.

Comments:

Did you use Fc blocker?

Figure 1. The does not show what denotes the three groups of participants.

Figure 4A and 5A. Medians for ART- are not visible.

Figure 6B: show correlation line.

Line 140: CD45 is also expressed on myeloid cells. Gating thorough CD45 results targeting PBMC and not just lymphocytes.

Line 247: what is MW?

Line 257: cytolytic, and not citolytic.

Round 2

Reviewer 1 Report

Comments and Suggestions for Authors

Authors have done all recommended changes.